# Reward Propagation
# Using Graph Convolutional Networks

**Martin Klissarov**
Mila, McGill University
martin.klissarov@mail.mcgill.ca

**Doina Precup**
Mila, McGill University and DeepMind
dprecup@cs.mcgill.ca

## Abstract

Potential-based reward shaping provides an approach for designing good reward functions, with the purpose of speeding up learning. However, automatically finding potential functions for complex environments is a difficult problem (in fact, of the same difficulty as learning a value function from scratch). We propose a new framework for learning potential functions by leveraging ideas from graph representation learning. Our approach relies on Graph Convolutional Networks which we use as a key ingredient in combination with the probabilistic inference view of reinforcement learning. More precisely, we leverage Graph Convolutional Networks to perform message passing from rewarding states. The propagated messages can then be used as potential functions for reward shaping to accelerate learning. We verify empirically that our approach can achieve considerable improvements in both small and high-dimensional control problems.

## 1 Introduction

Reinforcement learning (RL) algorithms provide a solution to the problem of learning a policy that optimizes an expected, cumulative function of rewards. Consequently, a good reward function is critical to the practical success of these algorithms. However, designing such a function can be challenging [Amodei et al., 2016]. Approaches to this problem include, amongst others, intrinsic motivation [Oudeyer and Kaplan, 2007, Schmidhuber, 2010], optimal rewards [Singh et al., 2010] and potential-based reward shaping [Ng et al., 1999]. The latter provides an appealing formulation as it does not change the optimal policy of an MDP while potentially speeding up the learning process. However, the design of potential functions used for reward shaping is still an open question.

In this paper, we present a solution to this problem by leveraging the probabilistic inference view of RL [Toussaint and Storkey, 2006, Ziebart et al., 2008]. In particular, we are interested in formulating the RL problem as a directed graph whose structure is analogous to hidden Markov models. In such graphs it is convenient to perform inference through message passing with algorithms such as the forward-backward algorithm [Rabiner and Juang, 1986]. This inference procedure essentially propagates information from the rewarding states in the MDP and results in a function over states. This function could then naturally be leveraged as a potential function for potential-based reward shaping. However, the main drawback of traditional message passing algorithms is that they can be computationally expensive and are therefore hard to scale to large or continuous state space.

We present an implementation that is both scalable and flexible by drawing connections to spectral graph theory [Chung, 1997]. We use Graph Convolutional Networks (GCN) [Kipf and Welling, 2016] to propagate information about the rewards in an environment through the message passing mechanism defined by the GCN's structural bias and loss function. Indeed, GCNs belong to the larger class of Message Passing Neural Networks [Gilmer et al., 2017] with the special characteristic that their message passing mechanism builds on the graph Laplacian. The framework of Proto-Value

Functions [Mahadevan, 2005] from the reinforcement learning literature has previously studied the properties of the graph Laplacian and we build on these findings in our work.

We first evaluate our approach in tabular domains where we achieve similar performance compared to potential based reward shaping built on the forward-backward algorithm. Unlike hand-engineered potential functions, our method scales naturally to more complex environments; we illustrate this on navigation-based vision tasks from the MiniWorld environment [Chevalier-Boisvert, 2018], on a variety of games from the Atari 2600 benchmark [Bellemare et al., 2012] and on a set of continuous control environments based on MuJoCo [Todorov et al., 2012] , where our method fares significantly better than actor-critic algorithms [Sutton et al., 1999a, Schulman et al., 2017] and additional baselines.

## 2 Background and Motivation

A Markov Decision Process $\mathcal{M}$ is a tuple $\langle \mathcal{S}, \mathcal{A}, \gamma, r, P \rangle$ with a finite state space $\mathcal{S}$, a finite action space $\mathcal{A}$, discount factor $\gamma \in [0, 1)$, a scalar reward function $r : S \times A \to Dist(\mathbb{R})$ and a transition probability distribution $P : S \times A \to Dist(S)$. A policy $\pi : S \to Dist(A)$ specifies a way of behaving, and its value function is the expected return obtained by following $\pi$: $V_\pi(s) \dot{=} \mathbb{E}_\pi \left[ \sum_{i=t}^{\infty} \gamma^{t-i} r(S_i, A_i) \big| S_t = s \right]$. The value function $V_\pi$ satisfies the following Bellman equation: $V_\pi(s) = \sum_a \pi(a|s) \left( r(s, a) + \gamma \sum_{s'} P(s'|s, a) V_\pi(s') \right)$ where $s'$ is the state following state $s$. The policy gradient theorem [Sutton et al., 1999a] provides the gradient of the expected discounted return from an initial state distribution $d(s_0)$ with respect to a parameterized stochastic policy $\pi_\theta$: $\frac{\partial J(\theta)}{\partial \theta} = \sum_s d(s; \theta) \sum_a \frac{\partial \pi(a|s)}{\partial \theta} Q_\pi(s, a)$ where we simply write $\pi$ for $\pi_\theta$ for ease of notation and $d(s; \theta) = \sum_{s_0} d(s_0) \sum_{t=0}^{\infty} \gamma^t P^\pi(S_t = s|S_0 = s_0)$ is the discounted state occupancy measure.

**Reward Shaping.** The framework reward shaping augments the original reward function by adding a shaping function, resulting in the following equation: $R'(S_t, A_t, S_{t+1}) = r(S_t, A_t) + F(S_t, S_{t+1})$ where $F(S_t, S_{t+1})$ is the shaping function which can encode expert knowledge or represent concepts such as curiosity [Schmidhuber, 2010, Oudeyer and Kaplan, 2007]. Ng et al. [1999] showed that a necessary and sufficient condition for preserving the MDP's optimal policy when using $R'$ instead of $r$ is for the shaping function to take the following form,

$$F(S_t, S_{t+1}) = \gamma \Phi(S_{t+1}) - \Phi(S_t)$$

where $\Phi$ is the scalar potential function $\Phi : S \to \mathbb{R}$, which can be any arbitrary function defined on states. In their work, the potential function was defined as a distance to a goal position. Different alternatives have been explored since, such as learning from human feedback [Harutyunyan et al., 2015] or using similarity-based shaping for learning from demonstrations [Brys et al., 2015]. Marthi [2007], Grzes and Kudenko [2010] have also considered automatically learning the potential function. These approaches either require a human in the loop or are not easily scalable to large problems. A key difference and contribution of this work is that we instead aim to learn end-to-end the potential function that scale naturally to complex environments.

**RL as Probabilistic Inference.** Consider the graphical model in Fig.1, where $O_t$ is a binary variable dependent on the action $A_t$ and the state $S_t$. The distribution over this optimality variable is defined with respect to the reward:

$$p(O_t = 1|S_t, A_t) = f(r(S_t, A_t))$$

where $f(r(S_t, A_t))$ is a function used to map rewards into probability space. In previous work [Toussaint, 2009, Levine, 2018], this is taken to be the exponential function (together with the assumption that rewards are negative) in order to facilitate derivation. In our case, we simply use the sigmoid function to allow for any types of rewards. In this model, the $O_t$ variables are considered to be observed while the actions $A_t$ and states $S_t$ are latent.

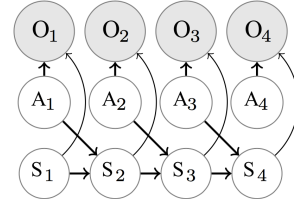

Figure 1: **Graphical model of the control task in RL.** $O_t$ is an observed variable, while $S_t$ and $A_t$ are the state and action latent variables.

This particular structure is analogous to hidden Markov models (HMM), where we can derive the forward-backward messages to perform inference [Ziebart et al.,

2008, Kappen et al., 2012, Toussaint and Storkey, 2006]. As we show next, in the case of reinforcement learning as a graphical model, message passing can be understood as inferring a function over states that is closely related to the value function.

As in the case of HMMs, the forward-backward messages in the reinforcement learning graph take following form, $\beta(S_t) = p(O_{t:T}|S_t)$ and $\alpha(S_t) = p(O_{0:t-1}|S_t)p(S_t)$ where the notation $O_{t:T}$ defines the set of variables $(O_t, O_{t+1}, ..., O_T)$. The backward messages $\beta(S_t)$ represent the probability of a future optimal trajectory given the current state, where future optimal trajectory can be translated as the probability of a high return given a state. This measure can be shown to be the projection into probability space of an optimistic variant of the value function from maximum-entropy RL [Levine, 2018]. In the case of the forward messages $\alpha(S_t)$, they represent the probability of a past optimal trajectory, scaled by the probability of the current state. There is currently no similar quantity in reinforcement learning, but a natural analogue would be to consider value functions that look backwards in time $V_\alpha^\pi(s) = \mathbb{E}[\sum_{i=0}^{t-1} \gamma^{t-i} r(S_i, A_i)|S_t = s]$.

By using the conditional independence properties of the RL graph, the terms for the forward and backward messages can be expanded to obtain their recursive form,

$$\alpha(S_t, A_t) = \sum_{S_{t-1}} \sum_{A_{t-1}} p(S_t|S_{t-1}, A_{t-1})p(A_t)p(O_{t-1}|S_{t-1}, A_{t-1})\alpha(S_{t-1}, A_{t-1}) \qquad (1)$$

$$\beta(S_t, A_t) = \sum_{S_{t+1}} \sum_{A_{t+1}} p(S_{t+1}|S_t, A_t)p(A_{t+1})p(O_t|S_t, A_t)\beta(S_{t+1}, A_{t+1}) \qquad (2)$$

With the associated base case form:

$$\beta(S_T, A_T) = f(r(S_T, A_T)), \quad \alpha(S_1, A_1) \propto \mathbb{E}_{S_0, A_0}[f(r(S_0, A_0))] \qquad (3)$$

Since potential based functions for reward shaping are defined only on the state space, we will marginalize the actions and use the marginalized messages defined as $\alpha(S_t)$ and $\beta(S_t)$. Given that the forward-backward messages carry meaningful quantities with respect to the rewards in an MDP, they are well-suited to be used as potential functions:

$$F(S_t, S_{t+1}) = \gamma \Phi_{\alpha\beta}(S_{t+1}) - \Phi_{\alpha\beta}(S_t) \qquad (4)$$

where $\Phi_{\alpha\beta}(S_t) = \alpha(S_t)\beta(S_t) \propto p(O_{0:T}|S_t)$. The potential function then represents the probability of optimality for a *whole* trajectory, given a state. That is, they represent the probability that a state lies in the path of a high-return trajectory. This again is in contrast to the usual value function which considers only future rewarding states, given a certain state.

Obtaining the messages themselves, however, is a challenging task as performing exact inference is only possible for a restricted class of MDPs due to its computation complexity of $O(N^2 T)$ where $N$ is the number of states and $T$ is the length of a trajectory. As we are interested in a generalizable approach, we will leverage the recently introduced Graph Convolutional Networks [Kipf and Welling, 2016, Defferrard et al., 2016].

**Graph Convolutional Networks.** Graph Convolutional Networks (GCNs) have mainly been used in semi-supervised learning for labeling nodes on a graph. The datasets' labeled nodes contain information that is propagated by the GCN, leading to a probability distribution defined over all nodes. A 2-layer GCN can be expressed as:

$$\Phi_{GCN}(X) = \text{softmax}\big(\hat{T} \text{ ReLU}\big(\hat{T} X W^{(0)}\big) W^{(1)}\big) \qquad (5)$$

where $W^{(i)}$ is the weight matrix of layer $i$ learned by gradient descent and $X$ is the input matrix with shape $N^{nodes} \times M^{features}$. The matrix $\hat{T}$ is a transition matrix defined as $\hat{T} = D^{-1/2}\tilde{A}D^{-1/2}$, where $A$ is the adjacency matrix with added self-connections and $D$ is the degree matrix, that is $D_{ii} = \sum_j A_{ij}$. At the core of the mechanism implemented by GCNs is the idea of spectral graph convolutions which are defined through the graph Laplacian. In the next section we highlight some of its properties through the Proto-Value Function framework [Mahadevan, 2005, Mahadevan and Maggioni, 2007] from the reinforcement learning literature.

## 3  Proposed Method

We propose to apply GCNs on a graph in which each state is a node and edges represent a possible transition between two states. In this graph we will propagate information about rewarding states

through the message passing mechanism implemented by GCNs. The probability distribution at the output of the GCN, defined as $\Phi_{GCN}(s)$, can then be used as a potential function for potential-based reward shaping.

To clearly define a message passing mechanism it is necessary to establish the base case and the recursive case. This is made explicit through the GCN's loss function,

$$\mathcal{L} = \mathcal{L}_0 + \eta \mathcal{L}_{prop} \tag{6}$$

where $\mathcal{L}_0$ is the supervised loss used for the base case and $L_{prop}$ the propagation loss implementing the recursive case. We define the base case similarly to the forward-backward algorithm, that is, by considering the first and last states of a trajectory. Additionally, as our goal is to propagate information from rewarding states in the environment, we emphasize this information by including in the set of base cases the states where the reward is non-zero. As shown in Eq.3, the value of the base case depends directly (or through expectation) on the environmental reward. When using GCNs, it is straightforward to fix the values of the base cases by considering its supervised loss, defined as the cross entropy between labels $Y$ and predictions $\hat{Y}$ which is written as $H(Y, \hat{Y})$. To implement the base case, the supervised loss then simply takes the following form,

$$\mathcal{L}_0 = H(p(O|S), \Phi_{GCN}(S)) = \sum_{S \in \mathbb{S}} p(O|S) \log \left( \Phi_{GCN}(S) \right)$$

where $\mathbb{S}$ is the set of base case states. The recursive case of the message passing mechanism is attended by the propagation loss in Eq.6 defined as,

$$\mathcal{L}_{prop} = \sum_{v,w} A_{vw} ||\Phi_{GCN}(X_w) - \Phi_{GCN}(X_v)||^2$$

where $A_{vw}$ is the adjacency matrix taken at node $v$ and $w$. While the recursive form of the forward-backward messages in Eq.1-2 averages the neighboring messages through the true transition matrix, the GCN's propagation loss combines them through the graph Laplacian. Moreover, the mechanism of recursion is also at the core of the GCN's structural bias. To see why, we have to consider them as a specific instance of Message Passing Neural Networks (MPNN) [Gilmer et al., 2017]. More precisely, the messages that GCNs propagate take the form: $m_v = \sigma \left( W^T \sum_w \hat{T}_{vw} m_w \right)$ where $W$ is the matrix of parameters, $v$ is a specific node, $w$ are its neighbours and $m_w$ is the message from node $w$. We now motivate the choice of using the graph Laplacian as a surrogate.

**Motivations for using the graph Laplacian.**    The main approximation introduced thus far was to replace the true transition matrix by the graph Laplacian. This approximation is also at the core of the Proto-Value Function framework [Mahadevan and Maggioni, 2007] which addresses the problem of representation learning using spectral analysis of diffusion operators such as the graph Laplacian. Proto-value functions are defined as the eigenvectors following the eigendecomposition of the true transition matrix $P^\pi$ of an MDP. These vectors can then be linearly combined to obtain any arbitrary value function. However, as the transition matrix is rarely available and hard to approximate, Mahadevan and Maggioni [2007] use the graph Laplacian matrix as an efficient surrogate to perform the eigendecomposition. This choice is motivated by the fact that projections of functions on the eigenspace of the graph Laplacian produce the smoothest approximation with respect to the underlying state-space topology of the MDP [Chung, 1997], where smoothness is defined through the Sobolov norm [Mahadevan and Maggioni, 2006]. As value functions are generally smooth and respect the MDP's underlying topology, the graph Laplacian is considered to be a viable surrogate. In our work, we do not aim at approximating the value function, but we argue that preserving these qualities is important in order to obtain a useful signal to accelerate learning.

## 3.1 Practical Considerations

To implement our approach on a wide range of MDPs, the question of how to estimate the underlying graph arises. In the Proto-Value Function framework, the authors draw a large number of samples and incrementally construct the underlying graph. However, representing the whole graph is not a scalable solution as performing inference (even on GPU) would be too costly. A possible solution would be to consider discretizing the state-space. Indeed, it has been shown that animals explore their environment with the help of grid cells that activate on particular intervals [O'Keefe and Dostrovsky, 1971, Moser

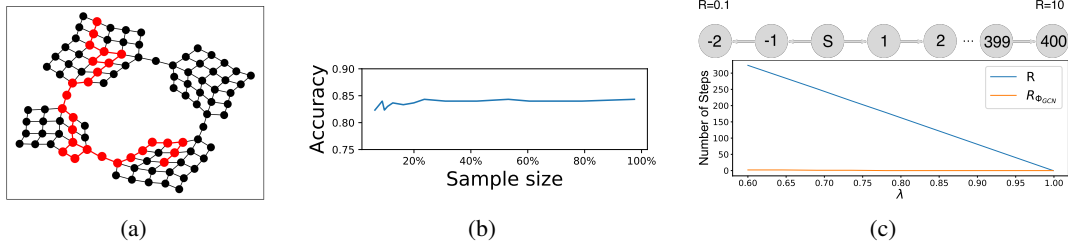

(a)                  (b)                               (c)

Figure 2: a) **Underlying graph for the FourRooms domain.** Red nodes and edges represent the sampled trajectory in the environment. b) **Validation accuracy on the Cora dataset.** We train a GCN on this dataset by only providing a certain percentage of the total training data at each iteration. Although we only provide sample graphs, the validation accuracy remains mostly constant. c) **Numbers of iterations to convergence** between the GCN-shaped reward function $R_{\Phi_{GCN}}$ and the original reward function $R$.

and Moser, 1998, Banino et al., 2018]. However, such an implementation would introduce a whole range of hyperparameters that would require tuning and might severely affect the final performance.

To address this, we propose a straightforward solution by choosing to approximate the underlying graph through sampled trajectories on which we train the GCN. The idea is illustrated in Fig 2a, where the whole graph is shown, but we have highlighted in red only a subset of states and edges corresponding to a sampled trajectory used to train the GCN at this particular iteration.

To investigate whether this sampling approach will still encompass the complexities of the whole graph, we apply the GCN on samples from the Cora dataset [Sen et al., 2008] and evaluate its validation accuracy on the whole graph, as presented in Fig.2b. We notice that although we only use samples of the whole graph, the GCN is still able to implicitly encode the underlying structure and maintain a similar validation accuracy across sample size. This is made possible by learning the weights of the GCNs, making this a paramater-based approach. An important benefit of this solution is the fact that the computational overhead is greatly reduced, making it a practical solution for large or continuous MDPs. We investigate further the consequences of this approximation in Appendix A.8 where we show that the resulting function on states, $\Phi(s)$, will tend to be more diffuse in nature. This function over states can then potentially be leveraged by the agent to learn more efficiently. Our sampling strategy is closely related to the one employed by previous work based on the graph Laplacian [Machado et al., 2017a,c], although with the important difference that we do not proceed to the eigen-decomposition of the transition matrix.

### 3.2 Illustration

To illustrate the benefits offered by forming potential based functions through the GCN's propagation mechanism, we consider a toy example depicted in Fig.2c where the agent starts in the state S. It can then choose to go left and obtain a small reward of 0.1 after 2 steps or go right where after 400 steps it gets a reward of 10. The optimal policy in case is to go right. In this toy example we update both actions at each iteration avoiding any difficulties related to exploration. The targets used to update the action-value function are the $\lambda$-returns. In Fig.2c we plot the number of iterations required to converge to the optimal policy as a function of the $\lambda$ parameter.

We notice that, in the case of the original reward function (denoted as $R$), the number of steps required to converge depends linearly on the value of $\lambda$, whereas the potential shaped reward function (denoted as $R_\Phi$) is mostly constant. The only value for which both methods are equal is when $\lambda = 1$. However, in practical settings, such high vales of $\lambda$ lead to prohibitively high variance in the updates. The observed difference between the two approaches has previously been investigated more rigorously by Laud and DeJong [2003]. The authors show that the number of decisions it takes to experience accurate feedback, called the *reward horizon*, directly affects how difficult it is to learn from this feedback. When using potential based reward shaping, the reward horizon can then be scaled down, reducing the learning difficulty. However, as our approach learns the potential function by experiencing external rewards, its purpose is not to improve exploration. Instead, our approach can be understood as an attempt to accelerate learning by emphasizing information about rewarding states through potential functions with the guarantee of preserving the optimal policy.

### 3.3 Algorithm

We now present our implementation of the ideas outlined above in the policy gradient framework.[1] We define two kinds of action value functions that will be used: the original function, $Q^\pi(s,a) = \mathbb{E}\big[\sum_t \gamma^t r(S_t, A_t)\big]$ and the reward-shaped function, $Q_\Phi^\pi(s,a) = \mathbb{E}\big[\sum_t \gamma^t(r(S_t, A_t) + \gamma\Phi_{GCN}(S_{t+1}) - \Phi_{GCN}(S_t))\big]$. We combine them though a scalar $\alpha$ as in $Q_{comb}^\pi = \alpha Q^\pi(s,a) + (1-\alpha)Q_\Phi^\pi(s,a)$. Algorithm 1 then describes the end-to-end training approach. Potential-based reward shaping in the context of policy-based approaches has interesting connections to the use of a baseline [Schulman et al., 2016]. In our approach, we can use the identity $Q^\pi(s,a) - \Phi(s) = Q_\Phi^\pi(s,a)$ to notice that the resulting baseline is simply the potential function at a given state. In general, it is hard to evaluate the benefits of using a particular baseline, although it is possible to obtain bounds on the resulting variance as shown in Greensmith et al. [2004]. Adopting their analysis, we show in Appendix A.7 that the highest upper bound is less or equal than the one obtained by the value function $V^\pi(s)$.

---

**Algorithm 1:** Reward shaping using GCNs

---

Create empty graph $G$
**for** *Episode=0,1,2,....* **do**
    **for** *t=1,2...T* **do**
        | Add transition $(S_{t-1}, S_t)$ to graph $G$
    **end**
    **if** *mod(Episode,N)* **then**
        | Train the GCN on the approximate graph.
    $Q_{comb}^\pi = \alpha Q^\pi + (1-\alpha)Q_\Phi^\pi$
    Maximize $E_\pi\big[\nabla\log\pi(A_t|S_t)Q_{comb}^\pi(S_t, A_t)\big]$
    Reset $G$ to empty graph (optional)
**end**

---

## 4 Experiments and Results

### 4.1 Tabular

*Experimental Setup:* We perform experiments in the tabular domains depicted in Fig. 3 where we explore the classic FourRooms domain and the FourRoomsTraps variant where negative rewards are scattered through the rooms. The results are presented in the form of cumulative steps (i.e. regret). In these experiments we add noise in the action space (there is a 0.1 probability of random action) in order to avoid learning a deterministic series of actions. We compare our approach, denoted $\Phi_{GCN}$, to a actor-critic algorithm using $\lambda$-returns for the critic, denoted A2C.

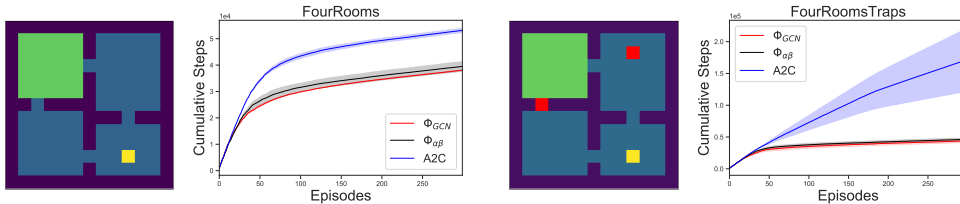

Figure 3: **Tabular environments.** Results are presented for the classic FourRooms domain and a variant called FourRoomsTraps. In both cases the agent starts in a random position inside the green square and has to get to the yellow goal, while avoiding red regions.

Our approach shows significant improvements over the classic actor-critic algorithm, which suggests that our method is able to provide the agent with valuable information to accelerate learning. As the adjacency matrix can be fully constructed, we also compare our approach to a formal implementation of the forward-backward algorithm in order to propagate reward information through the RL graph

(as in Eq. 4) denoted as $\Phi_{\alpha\beta}$. In Fig.3, we notice that both message passing mechanisms produce very similar results in terms of performance. We also provide illustrations of $\Phi_{GCN}$ and $\Phi_{\alpha\beta}$ in Appendix A.1 where we show very similar distributions over states and where we include all the values of the hyperparameters.

## 4.2 MiniWorld

*Experimental Setup:* We further study how we can improve performance in more complex environments where hand-designed potential functions are hard to scale. We work with the MiniWorld [Chevalier-Boisvert, 2018] simulator and explore a vision-based variant of the classic FourRooms domains called MiniWorld-FourRooms-v0 in which the agent has to navigate to get to the red box placed at a random position throughout the rooms. We also experiment with MiniWorld-MyWayHome-v0 which is the analogue of the challenging Doom-MyWayHome-v0 [Kempka et al., 2016]. The goal is to navigate through nine rooms with different textures and sizes to the obtain a reward. Finally, the last environment is MiniWorld-MyWayHomeNoisyTv-v0, a stochastic variant on MiniWorld-MyWayHome-v0 inspired by Burda et al. [2018a] that introduces a television activated by the agent that displays random CIFAR-10 images [Krizhevsky et al.]. In all environments we use Proximal Policy Optimization [Schulman et al., 2017] for the policy update. All details about hyperparameters and network architectures are provided in the Appendix A.2. However, it is important to note that through all these experiments we use the same values for the hyperparameters, making it a general approach. For all the experiments presented in Fig. 4 we add noise in the action space and randomize the agent's starting position to avoid deterministic solutions.

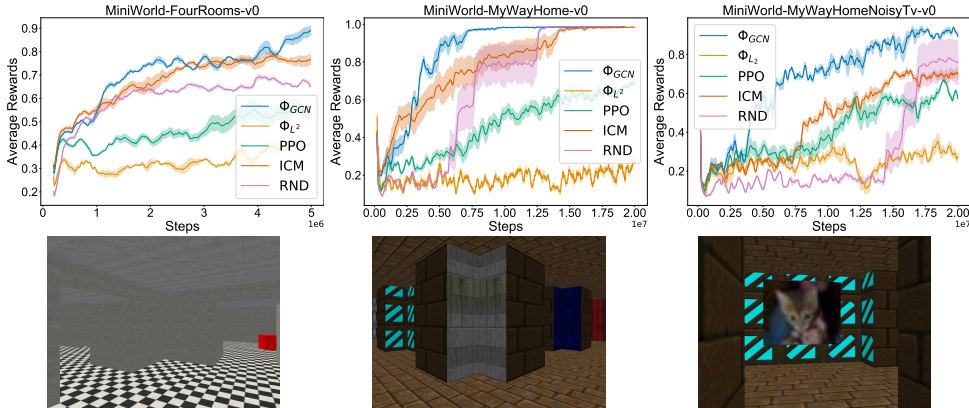

Figure 4: **High dimensional control.** In all environments, the results show the difficulty of constructing a helpful potential function through the $\Phi_{L^2}$ baseline. Our approach on the other hand almost doubles the PPO baseline. We also present other approaches using reward shaping where the aim is to improve exploration.

To highlight the difficulty of learning potential functions in the environments we investigate, we provide an additional baseline, $\Phi_{L^2}$, that naively implements the $L^2$ distance between a state and the goal and leverage it as the potential function. For completeness, we also provide two exploration baselines based on reward shaping: the random network distillation approach (RND) [Burda et al., 2018b] and the intrinsic curiosity module (ICM) [Pathak et al., 2017]. However, we emphasize that our approach is not meant to be a solution for exploration, but instead aims at better exploiting a reward signal in the environment.

The results in Fig. 4 show that in all environments, the naive reward shaping function does not help performance while our approach almost doubles the baseline score. We also notice that compared to the exploration baseline we learn faster or achieve better final performance. The greatest difference with exploration approaches is shown in the MiniWorld-MyWayHomeNoisyTv-v0 task, where exploration approaches are less robust to stochasticity in the environment. This again highlights a major difference of our approach: it does not necessarily seek novelty. Moreover, we emphasize that potential-based reward shaping is the only approach that guarantees invariance with respect to the optimal policy.

An important hyperparameter in our approach is effectively $\alpha$ which trades-off between the reward shaped return and the default return. In Fig 10 of the Appendix A.3 we show the results obtained on MiniWorld-FourRooms-v0 across all values of $\alpha$. We also investigate the role of $\eta$, the hyperparameter trading-off between the two losses of the GCN, in Appendix A.4 and provide evidence that it controls the bias-variance trade-off of the propagation process.

### 4.3 Atari 2600

*Experimental Setup:* The Atari benchmark is relevant to investigate as it includes a wide variety of games that range from reactive games to hard exploration games. We showcase the applicability of our method by running experiments over 40M frames on 20 Atari games that exhibit such variety. We include sticky actions to avoid deterministic solutions [Machado et al., 2017b] and use the same values of hyperparameters across all games (details are in Appendix A.5).

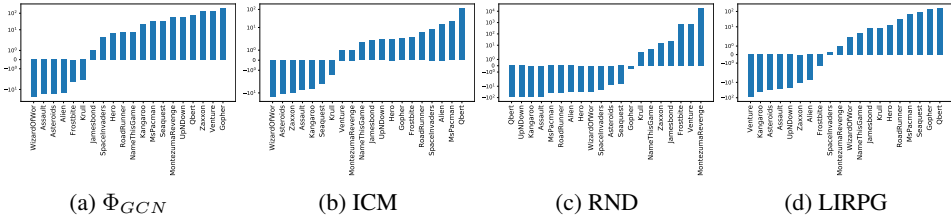

(a) $\Phi_{GCN}$      (b) ICM      (c) RND      (d) LIRPG

Figure 5: **Performance improvement** in log scale over the PPO baseline where we compare $\Phi_{GCN}$ to the intrinsic curiosity module (ICM), Random Network Distillation (RND) and Learning Intrinsic Rewards for Policy Gradient (LIRPG) .

As in the MiniWorld experiments, we compare our approach to two exploration baselines, that is ICM and RND, and present in Fig. 5 the percentage of improvement over the PPO algorithm. Additionally, we also compare with Learning Intrinsic Rewards for Policy Gradient (LIRPG) [Zheng et al., 2018]. This baseline is more closely related to our approach in the sense that it does not specifically aim to improve exploration but still seeks to improve the agent's performance. Note however that only our approach is guaranteed invariance with respect to the optimal policy as we are building on the potential-based reward shaping framework while LIRPG builds on the optimal rewards framework [Singh et al., 2010].

| Method | FPS |
|---|---|
| PPO | 1126 |
| $\Phi_{GCN}$ | 1054 |
| RND | 987 |
| ICM | 912 |
| LIRPG | 280 |

Table 1: Frames-Per-Second (FPS) on Atari games

We notice that in most games $\Phi_{GCN}$ shows good improvements especially in games such as Gopher or Zaxxon. Moreover, we notice that our approach is more consistent in improving the policy's performance as compared to the other approaches. Interestingly, the RND approach provides great gains on hard exploration games such as Venture, but otherwise reduces dramatically the score in a range of games. Finally, in Table 1 we also compare the run-time of all the presented approaches. We did these evaluations on a single V100 GPU, 8 CPUs and 40GB of RAM. The time taken, in frames-per-second (FPS), for our approach $\Phi_{GCN}$ is very similar to the PPO baseline, only slightly slower. We also compare favourably with respect to the RND, ICM, and LIRPG baselines. We believe this good performance stems directly from our sampling strategy that is minimal yet effective.

### 4.4 MuJoCo

*Experimental Setup:* To further investigate the applicability of our method, we perform experiments on environments with continuous states and actions by working with the MuJoCo benchmark [Todorov et al., 2012]. To make the learning more challenging, we experiment with the delayed version of the basic environments where the extrinsic reward is rendered sparse by accumulating it over 20 steps before it is being provided to the agent. All values of hyperparameters are provided in Appendix A.6.

In Fig. 6 we witness that our approach still provides significant improvements over the PPO baseline. We also compare to the LIRPG approach and notice that although $\Phi_{GCN}$ seems to learn a bit slower in the first iterations, the final performance presents a clear picture. We do not provide a comparison

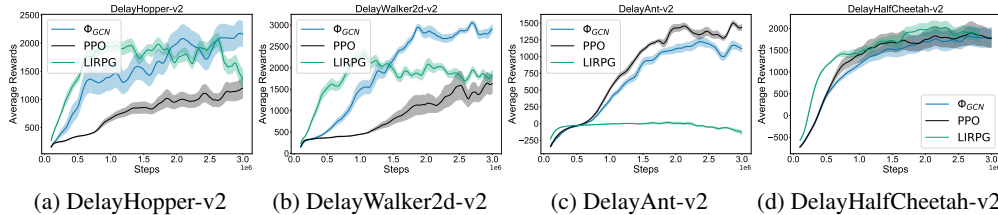

| (a) DelayHopper-v2 | (b) DelayWalker2d-v2 | (c) DelayAnt-v2 | (d) DelayHalfCheetah-v2 |

Figure 6: **Continuous control.** In most environments we see significant improvements over the PPO baseline. We also compare favorably to the Learning Intrinsic Rewards for Policy Gradient (LIRPG) baseline.

to ICM and RND as the former has been formulated for discrete actions while the latter was designed to receive pixels as inputs.

## 5 Related Work

Machado et al. [2017a,c] extend the Proto-Value Functions framework to the hierarchical reinforcement learning setting. The eigenvectors of the graph Laplacian are used to define an intrinsic reward signal in order to learn options [Sutton et al., 1999b]. However, their reward shaping function is not guaranteed invariance with respect to the original MDP's optimal policy. More recently, Wu et al. [2018] propose a way to estimate the eigendecomposition of the graph Laplacian by minimize a particular objective inspired by the graph drawing objective [Koren, 2003]. The authors then propose to use the learned representation for reward shaping but restrict their experiments to navigation-based tasks. Our approach is also reminiscent of the Value Iteration Networks [Tamar et al., 2016] where the authors propose to use convolutional neural networks to perform value iteration. In their approach, value iteration is performed on an approximate MDP which may not share similarities to the original MDP, whereas we propose to approximate the underlying graph of the original MDP. Return Decomposition for Delayed Rewards [Arjona-Medina et al., 2018] is another reward shaping method that tackles the problem of credit assignment through the concept of reward redistribution. However, it is not straightforward to apply in practice whereas potential-based reward shaping takes a simple form. Finally, our method is most closely related to the DAVF approach [Klissarov and Precup, 2018], however in their work the authors leverage GCNs to obtain value functions whereas we look to define potential functions for reward shaping.

## 6 Discussion

We presented a scalable method for learning potential functions by using a Graph Convolutional Network to propagate messages from rewarding states. As in the Proto-Value Function framework [Mahadevan and Maggioni, 2007, Mahadevan, 2005] this propagation mechanism is based on the graph Laplacian which is know to induce a smoothing prior on the functions over states. As shown in the experiments and illustrations, the resulting distribution over states can then be leveraged by the agent to accelerate the performance. Moreover, unlike other reward shaping techniques, our method is guaranteed to produce the same optimal policy as in the original MDP.

We believe that our approach shows potential in leveraging advances from graph representation learning [Bronstein et al., 2016, Hamilton et al., 2017] in order to distribute information from rewarding states. In particular it would be interesting to explore with transition models other than the graph Laplacian that would be more closely related to the true transition matrix $P^\pi$, as in done in Petrik [2007] by the introduction of Krylov bases. Another possible improvement would be to represent actions and potentially the policy itself when approximating the graph. This could be leveraged through look-ahead advice [Wiewiora et al., 2003].

Finally, in this paper we moslty took a model-free approach to exploit GCNs in the context of reinforcement learning. Future work could consider a more sophisticated approach by taking inspiration form grid cell-like constructs [O'Keefe and Dostrovsky, 1971] or by combining our sampling strategy with model roll-outs [Sutton, 1991] to construct the MDP's underlying graph.

## Broader Impact

We believe that our research provides scalable ways to learn potential functions for reward shaping yet maintaining guarantees of invariance with respect to the optimal policy of the original setting. We believe that one of the most promising attribute of our line of research is the fact that the optimal policy remains unchanged. This is a fundamental feature for sensitive applications such as healthcare, recommendation systems and financial trading.

Moreover, by using potential based reward shaping, our method is meant to accelerate learning. This is another very important characteristic with regards to applications where sample complexity, memory and compute are of importance. Accelerating learning can also have downsides in the situations where there is competition between technologies. This could lead to one competitor obtaining an advantage due to faster learning, which can then exacerbate this advantage (rich get richer situation). We suggest that any publications that proceed in this line of research to be open about implementation details and hyperparameters.

Another point to consider when proceeding to applications is related to the complexity of the MDP. If the application is small enough, it would be a good approach to try and store the whole underlying graph. We expect that in such settings our approach will provide significant improvements with reduced complexity when compared to approaches that require the eigen-decomposition of a transition model. If the MDP is too large to be reconstructed, we suggest applying our sampling strategy that will avoid increases in computational complexity. In these settings, we believe our approach can still provide robust improvements, however it might be more sensitive to the choice of hyperparameters. However, we have shown that across a wide range of games, high values of $\alpha$ provide good improvements and we expect this to be the case in many applications.

## Acknowledgments and Disclosure of Funding

The authors would like to thank the National Science and Engineering Research Council of Canada (NSERC) and the Fonds de recherche du Quebec - Nature et Technologies (FRQNT) for funding this research; Khimya Khetarpal for her invaluable and timely help; Zafareli Ahmed and Sitao Luan for useful discussions on earlier versions of the draft and the anonymous reviewers for providing critical and constructive feedback.

## Footnotes

[1]The ideas are general and could also be used with Q-learning or other policy-based methods.

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
