[Supplementary Material]

# A Appendix

## A.1 Tabular experiments

### A.1.1 Implementation Details

For our experiments of the FourRooms and FourRoomsTraps domains we based our implementation on [Bacon et al., 2016] and ran the experiments for 300 episodes that last a maximum of 1000 steps. As these are tabular domains, each state is defined by a single feature for both the actor and the critic. For the $\Phi_{GCN}$ and $\Phi_{\alpha\beta}$ results we used the full graph of the MDP to compute the message passing mechanism. Full hyperparameters are listed here:

| Hyperparameter | Value |
|---|---|
| Actor lr | 1e-1 |
| Critic lr | 1e-1 |
| Discount | 0.99 |
| Max Steps | 1000 |
| Temperature | 1e-1 |
| GCN: hidden size | 64 |
| GCN: $\alpha$ | 0.6 |
| GCN: $\eta$ | 1e1 |

Table 2: Hyperparameters for the FourRooms and FourRoomsTraps domain

### A.1.2 Visualization

We visually inspect the output of $\Phi_{GCN}$ and $\Phi_{\alpha\beta}$ on the FourRooms and FourRoomsTraps domain. In Fig we notice a close ressemblance in the final output. In this particular environment, performing the messages obtained the forward-backward algorithm are therefore well approximated by the propagation mechanism of the GCN. This also results in very similar empirical performance.

(a) FourRooms       (b) $\Phi_{\alpha\beta}$       (c) $\Phi_{GCN}$

Figure 7: Visualization of the function over states obtained through message passing $\Phi_{\alpha\beta}$ and by the GCN $\Phi_{GCN}$ on the FourRooms domain where the agent starts in the upper left states (green) and has to get to the goal (yellow) in the bottom right room.

(a) FourRoomsTraps
(b) $\Phi_{\alpha\beta}$
(c) $\Phi_{GCN}$

Figure 8: Visualization of the function over states obtained through message passing $\Phi_{\alpha\beta}$ and by the GCN $\Phi_{GCN}$ on the FourRooms domain where the agent starts in the upper left states (green) and has to get to the goal (yellow) in the bottom right room while aboid the negative rewards (red).

Figure 9: Visualization of $\Phi_{GCN}$ across timesteps in an episode on the MiniWorld-FourRooms-v0 domain. **A** The agent is scanning the room in search of the red box (the goal). **B** While scanning the room, the agent is faced with a close wall, therefore the output of the GCN is low as there is nothing promising at this timestep. **C** The agent is about to enter the hallway between two rooms. We notice that consistently the output of the GCN spikes at such key moments. **D** The agent is about to cross to the next room and the output of the GCN is high to encourage crossing over. **E** The agent has seen the red box and the GCN's output spikes in order to push the agent towards the goal.

## A.2 High-Dimensional control

For experiments on MiniWorld-FourRooms-v0, MiniWorld-MyWayHome-v0 and MiniWorld-MyWayHomeNoisyTv-v0 we based our implementation on [Kostrikov, 2018] and ran the experiments for 5M steps and 20M steps respectively with 10 random seeds. The input provided to the GCN is the last hidden layer of the shared CNN network used by the actor and the critic. The architecture for the actor-critic was kept to be the same with respect to the original codebase. We provide a full list of the values for the hyperparameters:

We also provide visualization of the output of the GCN on the MiniWorld-FourRooms-v0 domain, as shown in Figure 9. In general, we notice that the output of the GCN is low in uninteresting moments (such as the agent facing the wall), but increases at key moments such as when it crosses hallways between rooms and in the sight of the goal.

Finally, we provide the hyperparameters used for experiments in Table 2.

| Hyperparameter | Value |
|---|---|
| Learing rate | 2.5e-4 |
| $\gamma$ | 0.99 |
| $\lambda$ | 0.95 |
| Entropy coefficient | 0.01 |
| LR schedule | constant |
| PPO steps | 128 |
| PPO cliping value | 0.1 |
| # of minibatches | 4 |
| # of processes | 32 |
| GCN: $\alpha$ | 0.8 |
| GCN: $\eta$ | 1e1 |
| ICM coeff. | 1e-2 |
| RND coeff. | 1.0 |

Table 3: Hyperparameters for the MiniWorld experiments

## A.3 Sweep across values of $\alpha$

In Fig.10 we present a sweep over the values of $\alpha$ which controls the trade-off between the reward-shaped value function and the regular value function. We notice that improvements are consistent for values of $\alpha$ higher than 0.5.

Figure 10: Performance on MiniWorld-FourRooms-v0 across values of $\alpha$. Higher values lead to better results.

## A.4 Investigation of the hyperparameter $\eta$

(a) SMaze-v0 task.   (b) $\eta = 0.1$   (c) $\eta = 1.0$   (d) $\eta = 10.0$

Figure 11: **Comparison of the output** of $\Phi_{GCN}$ for different values of $\eta$ from Eq. 6, which controls the complexity of the output. The visualization is performed on the SMaze-v0 environment shown in Fig.11a where the agent is the blue dot starting in lower left corner and the goal in the green dot in the upper right corner. Lower values of $\eta$ lead to less propagation and therefore a more simple, yet biased, output. As $\eta$ increases, the output shows a more complex structure.

We investigate the effect of the hyperparameter $\eta$ in the GCN's loss function (Eq. 6) which mediates between $\mathcal{L}_0$ and $\mathcal{L}_{prop}$, where the former is the supervised loss and the latter the propagation loss. The latter depends on the adjacency matrix of the graph and can therefore accentuate the recursive mechanism of the GCN as opposed to the supervised loss which controls the base case values. The Fig. 11 illustrates the result of varying $\eta$ in the SMaze-v0 environment shown in Fig. 11a where the agent (blue dot) has to navigate to the goal (green dot). In Fig. 11b, we see that for a low value of $\eta$, the GCN outputs a simple solution where the states on the left have a low value (blue) while states on right have a higher value (yellow), which matches the position of the goal and the starting states. This

solution shows more bias than variance as it favors a simple output. As we increase the value of $\eta$, the propagation loss has greater importance and the output $\Phi_{GCN}$ shows more complexity, incorporating more information about the environment's dynamics, such as the walls and the specific position of the goal. There exists an interesting parallel with the $\lambda$ parameter in the TD($\lambda$) algorithm: $\lambda$ controls the trade-off between bias and variance of the return estimation, while $\eta$ controls the bias-variance trade-off of the propagation process.

## A.5 Atari 2600

For experiments on the Atari games we based our implementation on [Dhariwal et al., 2017] and we ran the experiments for 40M frames over 10 random seeds. The input provided to the GCN is the last hidden layer of the shared CNN network used by the actor and the critic. The architecture for the actor-critic was kept to be the same with respect to the original codebase. We provide a full list of the values for the hyperparameters:

| Hyperparameter | Value |
|---|---|
| Learing rate | 2.5e-4 |
| $\gamma$ | 0.99 |
| $\lambda$ | 0.95 |
| Entropy coefficient | 0.01 |
| LR schedule | constant |
| PPO steps | 128 |
| PPO cliping value | 0.1 |
| # of minibatches | 4 |
| # of processes | 8 |
| GCN: $\alpha$ | 0.9 |
| GCN: $\eta$ | 1e1 |
| ICM coeff. | 1e-2 |
| LIRPG coeff. | 0.01 |

Table 4: Hyperparameters for the Atari 2600 experiment used for the PPO, ICM and $\Phi_{GCN}$ algorithms.

| Hyperparameter | Value |
|---|---|
| Learing rate | 1e-4 |
| $\gamma$ | 0.99 |
| $\lambda$ | 0.95 |
| Entropy coefficient | 0.001 |
| LR schedule | constant |
| PPO steps | 128 |
| PPO cliping value | 0.1 |
| # of minibatches | 4 |
| # of processes | 128 |
| RND int. coeff. | 1.0 |
| RND ext. coeff. | 2.0 |

Table 5: Hyperparameters for the Atari 2600 experiment used for the RND algorithm. Notice that these hyperparameters are taken directly from the paper as experiments were performed on Atari.

We also provide a visualization of the output of the GCN in Figure 12. In general, we notice that the GCN is sensitive to the volume of oxygen in the tank: it will encourage an agent with a full tank to move downwards and an agent with an almost empty tank to move upwards. The output of the GCN is also sensitive to situations when the agent gets dangerously close to other submarines.

Figure 12: Visualization of the output of the GCN during an episode of the game SeaquestNoFrameSkip-v4. **A** The agent is waiting to fill up its oxygen tank, as plunging underwater before having a full tank is a direct loss of life. **B** The oxygen tank is full and the agent is moving towards the bottom of the sea **C** The agent stays at the most bottom part of the sea as this place is the safest. Indeed, if it moves up it has to deal with other submarines that shoot projectiles, as shown in the screenshot of the game. **D** The agent comes to close contact with another submarine, which would result in a loss of life, causing the GCN to output a decreased value. **E** The oxygen tank is getting low, which eventually leads to a loss of life. The GCN is gradually decreases its output as it can predict the incoming danger. **F** The agent momentarily moves up, which is the right thing to do in this situation: if it gets to the surface it can recharge its oxygen tank. **G** The agent has failed to reach the surface and loses a life.

## A.6 MuJoCo

For the MuJoCo experiments, we based our implementation on [Dhariwal et al., 2017] and we ran the experiments for 3M steps over 10 random seeds. The input provided to the GCN is the last hidden layer of the actor's MLP network. The architecture for the actor-critic was kept to be the same with respect to the original codebase. We provide a full list of the values for the hyperparameters:

| Hyperparameter | Value |
|---|---|
| Learing rate | 3e-4 |
| $\gamma$ | 0.99 |
| $\lambda$ | 0.95 |
| Entropy coefficient | 0.0 |
| LR schedule | constant |
| PPO steps | 2048 |
| PPO cliping value | 0.1 |
| # of minibatches | 32 |
| # of processes | 1 |
| GCN: $\alpha$ (Walker and Ant) | 0.6 |
| GCN: $\alpha$ (Hopper and HalfCheetah) | 0.7 |
| GCN: $\eta$ | 1e1 |
| LIRPG coeff. | 0.01 |

Table 6: Hyperparameters for the MuJoCo experiment

Figure 13: Results on 20 Atari games using sticky actions [Machado et al., 2017b]

## A.7 Intepretation as a baseline

In the potential-based reward shaping framework [Ng et al., 1999], it is shown that the optimal policy will remain invariant to the choice of potential function. This can easily be seen by considering the following relation,

$$Q^\pi(s, a) - \Phi(s) = Q_\Phi^\pi(s, a)$$

where $\Phi(s)$ is the reward shaping function, $Q^\pi(s, a)$ is the policy's action value function and $Q_\Phi^\pi(s, a)$ is the action value function with respect to the potential based shaped reward. When used in the context of policy-based methods, rewards shaping can be considered as introducing a baseline. In particular, if we considered the action value function obtained by combining the original reward and shaped reward through the mixing coefficient $\alpha$, that is $Q_{comb}^\pi = \alpha Q^\pi(s, a) + (1 - \alpha)Q_\Phi^\pi(s, a)$, we obtain

$$\frac{\partial J(\theta)}{\partial \theta} = \sum_s d(s; \theta) \sum_a \frac{\partial \pi(a|s)}{\partial \theta} (Q_\pi(s, a) - (1 - \alpha)\Phi(s))$$

To investigate the consequences of such a baseline, we consider its effect on the variance of the policy gradient. Using results from Greensmith et al. [2004] (in particular Lemma 9), we denote $\text{Var}(\Phi)$ the variance of the estimate when using $(1 - \alpha)\Phi$ as a baseline and we obtain the following upper bound,

$$\text{Var}(\Phi) = \text{Var}(G) + \mathbb{E}\left[(1 - \alpha)^2 \Phi(s)^2 \mathbb{E}[(\frac{\partial J(\theta)}{\partial \theta})^2|s] - 2(1 - \alpha)\Phi(s)\mathbb{E}[(\frac{\partial J(\theta)}{\partial \theta})^2 Q_\pi(s, a)|s]\right]$$

where $\text{Var}(G)$ is the variance obtained by not using any baselines, that is the variance of the return $G$. When compared to the baseline defined as the state value function $V^\pi(s)$ (and denoted $\text{Var}(V^\pi)$), we can show that the highest upper bound will be less or equal.

Let's first consider the highest upper bound for the state value function. To do this we have to consider two cases. In the case where the highest reward, $R_{max}$, is smaller in magnitude than the lowest reward, $R_{min}$, (that is $|R_{max}| \leq |R_{min}|$) we obtain the following highest upper bound for the state value function,

$$\text{Var}(V^\pi) \leq \text{Var}(G) + \mathbb{E}\left[((\frac{R_{max}}{1 - \gamma})^2 - 2\frac{R_{max}R_{min}}{(1 - \gamma)^2})\mathbb{E}[(\frac{\partial J(\theta)}{\partial \theta})^2|s]\right] \text{ for } |R_{max}| \leq |R_{min}| \quad (7)$$

In the case where the highest reward, $R_{max}$, is greater in magnitude than the lowest reward, $R_{min}$, (that is $|R_{max}| \geq |R_{min}|$) we obtain the following highest upper bound for the state value function,

$$\text{Var}(V^\pi) \leq \text{Var}(G) + \mathbb{E}\left[((\frac{R_{min}}{1 - \gamma})^2 - 2\frac{R_{min}R_{max}}{(1 - \gamma)^2})\mathbb{E}[(\frac{\partial J(\theta)}{\partial \theta})^2|s]\right] \text{ for } |R_{min}| \leq |R_{max}| \quad (8)$$

We notice that these bounds are equal when $|R_{max}| = |R_{min}|$ and are the lowest when both values are low, which suggest that normalizing the rewards (and the returns) is a good general strategy for reducing variance. When using potential based reward shaping, we obtain the following upper bound irrespective of the magnitude of $R_{max}$ and $R_{min}$,

$$\text{Var}(\Phi) \leq \text{Var}(G) + \mathbb{E}\left[((\frac{(1 - \alpha)\sigma(R_{max})}{1 - \gamma})^2 - 2\frac{(1 - \alpha)\sigma(R_{max})R_{min}}{(1 - \gamma)})\mathbb{E}[(\frac{\partial J(\theta)}{\partial \theta})^2|s]\right] \quad (9)$$

where $\sigma$ is the sigmoid function. By inspection, we verify that for values of $R_{max}/(1 - \gamma)$ greater than $(1 - \alpha)\sigma(R_{max})$, the highest upper bound on the variance is strictly lower in the case of potential based reward shaping as defined by our approach when compared to either bounds obtained for the value function baseline (Eq.7-8). This is satisfied by almost all practical applications. Moreover, we notice that $\alpha$ controls the highest upper bound. Therefore, higher values of $\alpha$ are preferred in order to limit the possibility of higher variance.

## A.8 Approximating the adjacency matrix

Reconstructing the adjacency matrix is only possible if one tries to explicitly represent the underlying graph. When faced with high-dimensional MDPs, this is simply impractical. As pointed by Machado et al. [2017a], there is a trade-off representing accuractely the adjacency matrix and opting for a

simpler model through the incidence matrix, denoted as $C$. In their work, Machado et al. [2017a] show that by estimating the incidence matrix, they can recover the same eigenvectors as the ones obtained from the graph Laplacian. In our case, as we are not aiming to recover the eigenvectors of the graph Laplacian, we instead look at the consequence of using the incidence matrix as a transition operator for message passing.

We begin by noticing that the incidence matrix is in fact the un-normalized random walk matrix, that is $P^{rand} = D^{-1}C$ where $P^{rand}$ is the normalized random walk matrix and $D$ is the degree matrix. We show that the entropy rate of $P^{rand}$ is higher or equal when compared to the adjacency matrix or the true transition matrix $P^{\pi}$ for any policy $\pi$.

We start by providing the definition of entropy rate for the special case where the stochastic process is a Markov Chain $M$:

$$H(M) = -\sum_{ss'} \mu_s P_{ss'} \log P ss'$$

where $\mu_s$ is the stationary distribution of the arbitrary policy $\pi$ taken at state $s$. As we don't have access to this distribution, we will look at each of the rows of the matrices. We define a random walk matrix which preserve the original MDP's connectivity between states, that is, each non-zero in the original transition matrix $P$ is also a non-zero entry in the random walk matrix $P^{rand}$. We note that the entropy of a multinomial distribution is maximized when each entry is equal to $1/N$, where $N$ is the total number of entries. As each non-zero entry of the rows of matrix $P^{rand}$ contains such value, the entropy of any row in $P^{rand}$ is higher or equal than any row in any transition matrix $P$,

$$-\sum_{s'} P_{ss'} \log P_{ss'} \leq -\sum_{s'} P_{ss'}^{rand} \log P_{ss'}^{rand} \quad \forall \ i, P$$

We also notice that $\sum_s \gamma_s \sum_{s'} P_{ss'}^{rand} \log P_{ss'}^{rand} = \sum_{s'} P_{ss'}^{rand} \log P_{ss'}^{rand}$ as $\gamma$ is a probability vector that sums to 1, leading to a convex combination. We finally get that,

$$-\sum_s \mu_s \sum_{s'} P_{ss'} \log P_{ss'} \leq -\sum_{s'} P_{ss'}^{rand} \log P_{ss'}^{rand}$$

$$-\sum_s \mu_s \sum_{s'} P_{ss'} \log P_{ss'} \leq -\sum_s \gamma_s \sum_{s'} P_{ss'}^{rand} \log P_{ss'}^{rand}$$

The vector $\gamma$ is any vector summing to 1 and as such can be taken as the stationary distribution of the random walk matrix $P^{rand}$. As a direct consequence, the entropy rate of any MDP is less or equal to the entropy rate of the equivalent MDP where we have replaced the transition matrix with the random walk matrix $P^{rand}$. The derivation for the adjacency matrix follows the same intuition.

As the entropy rate gets higher, the complexity of the stochastic process and the number of probable paths from one state to any other state will increase. Recall that the GCN's loss function $\mathcal{L}_{prop}$ implements the recursive operation of message passing.

Intuitively, this means that the resulting distribution over states, $\Phi$, will be tend to be more diffuse in nature than the one obtained by true transition matrix or the adjacency matrix. This diffused signal can then be leveraged by the agent to learn more efficiently. However, it is important to notice that this does not guarantee better exploration, but instead an assignment mechanism that will produce a smoother signal, which the agent can learn to exploit. This also highlights a natural drawback previously observed in the PVF framework [Mahadevan, 2005]. In using diffusion operators the dependency on the policy $\pi$ is ignored. However, the potential function $\Phi(s)$ proposed in this work is not meant to replace the value function but instead to complement it in order to accelerate learning.