[Reviews · NeurIPS 2020]

Review 1

Summary and Contributions: This paper proposes a new framework for learning potential functions by leveraging ideas from graph representation learning. More precisely, it leverages Graph Convolutional Networks (GCN) to perform message passing over trajectory. So the potential functions for reward shaping from GCN can accelerate learning. The empirical results show that the approach can achieve considerable improvements in both small and high-dimensional control problems.

Strengths: The main strength of the work is to use GCN to approximate the potential functions for reward shaping.

Weaknesses: After reading the paper, it is hard to understand this paper in a high level, except its key topic is to accelerate learning using reward shaping (GCN is applied here). In the high level, does this paper only focus on inference? or both inference and learning? If this paper only use GCN in the inference stage, then its contribution is limited. Specifically, we know best P(O|S) (which is given) in Eq. 6, and then we can train GCN by minimizing Eq. 6. If we need to infer P(O|S) from RL, and in turn learn GCN parameters, then it is a challenge, the algorithm should not be easy as it stated in the paper. Moreover, the time complexity is much high because we need to count both GCN and RL steps.

Correctness: A little confused after reading the paper. Refer above.

Clarity: Yes. The paper may need to be reorganized (and it will be easy to understand by just saying for example training GCN with O given).

Relation to Prior Work: Yes

Reproducibility: Yes

Additional Feedback: The author needs to clarity the topic in a much high level. If this paper only applies GCN in RL, or GCN is pre-trained, its contribution is limited.


Review 2

Summary and Contributions: This paper proposes to learn the potential rewarding function by graphical convolutional network (GCN), in which each node indicates a state and an edge represents a possible transition between two states. The GCN loss contains two parts including a base supervised part and a message propagating part using the adjacency matrix. Practically, sampled trajectories / state transitions are used for training the GCN and the authors empirically show that such a choice does not affect the accuracy much. Experiments are conduct on a number of applications and the comprehensive results sufficiently support the proposed method.

Strengths: Encoding the states and state transitions as nodes and edges in a GCN is interesting. Experiments are conduct on a number of applications and the comprehensive results sufficiently support the proposed method.

Weaknesses: I only have one major concern that for many complex environments, the states are continuous and are not practical to be represented as nodes in a graph. Under such case, the state space is very large and sampling rollouts might not be adequate, and the accuracy might loss at a large margin. Could the author provide some possible solutions for this? Some minor problems: Line 76: missing a bracket for the function f() Line 78: “to to be” -> “to be” Line 215: use a different symbol instead of alpha which is used in Section 2 with a different meaning. I cannot find Algorithm 1 until I see the appendix, please move algorithm 1 in the main text or give a reference there. After rebuttal: The authors provided additional results to demonstrate that the method works well in continuous environment and I do not have other questions. I insist my original rating as an accept.

Correctness: Correct

Clarity: Well written

Relation to Prior Work: Clearly discussed

Reproducibility: Yes

Additional Feedback:


Review 3

Summary and Contributions: This paper uses Graph Convolutional Networks to propagating messages from rewarding states to non-rewarding ones, leading to better performance across a variety of domains. More specifically once the GCN passes information from rewarding states, the probability distribution may be used as a form of potential-based reward shaping. The efficacy of this method is evaluated on several different domains including gridworlds, 3-D navigation environments, and Atari games. Across these different settings, the method performs well when compared to A2C, PPO, ICM and RND.

Strengths: I thought this paper performed and admirable job of applying new techniques like GCN to at least partially address some of the difficulties of reward propagation and credit assignment in reinforcement learning. Overall, I due to the extensive experimental results and comparisons to other SOTA baselines, I have confidence that the algorithm is not only interesting, but also scalable and applicable to at least reasonably complex simulated environments.

Weaknesses: It would be interesting to understand the additional computational complexity introduced by this method (if any) compared to the baselines. I was unable to find any reports of wall-clock speed of the different implementations.

Correctness: Yes, as far as I can tell.

Clarity: Yes.

Relation to Prior Work: This paper may benefit from citing other work that similarly has looked into how to decompose rewarding states - one example would be RUDDER: Return Decomposion for Delayed Rewards (Arjona-Medina et al).

Reproducibility: Yes

Additional Feedback: Having read the rebuttal and (despite my attempts to the contrary) given the total absence of reviewer discussion on this paper. I'm sticking with my original score of 7.


Review 4

Summary and Contributions: The authors proposed a Graph Convolution Network (GCN) based potential function learning method for reward shaping, aiming at improving the policy learning speed. To avoid representing the whole transition graph, they adopted a sampling based approach that enables potential function learning on sampled trajectories with GCN. Their proposed method was evaluated on multiple domains, and the empirical results show that they improve over baseline policy learning method (PPO) and some exploration-encouraging methods (ICM and RND).

Strengths: - The authors proposed a scalable reward potential function learning method. Similar to previous work, the proposed algorithm utilizes sampled trajectories to estimate the MDP transition dynamics, avoiding the heavy computational overhead in maintaining a whole graphical representation for complex tasks. - The proposed algorithm aims at addressing the empirical convergence speed of reinforcement learning algorithms, which could be of great interests to the community. - The proposed method has achieved good empirical improvement over PPO on multiple domains. It also outperformed the ICM and RND approaches.

Weaknesses: - On the advantage and necessity of GCN. The method adapts a sampled trajectory-based approximation of the transition graphs. But given the trajectory samples, sequential models (RNN etc.) are sufficient to estimate the potential functions. It would be good if the authors can clarify the advantage and the necessity of GCN on the sampled trajectory inputs, compared to sequential models. - On the empirical baselines. The baselines, ICM and RND, are motivated to address hard exploration RL tasks, while the potential based reward shaping is motivated for faster convergence. They are related but address different issues. A more informative empirical comparison would be against the LIRPG (Learning Intrinsic Rewards for Policy Gradient from [a]), because both this paper and LIRPG aim at learning reward shaping for speed up policy learning. - On the applicability on continuous control problems. It would be good if the authors could discuss the applicability of the proposed method on continuous control problems. [a] Zheng, Zeyu, Junhyuk Oh, and Satinder Singh. "On learning intrinsic rewards for policy gradient methods." Advances in Neural Information Processing Systems. 2018. =========================== The responses addressed my concerns. The new additional results on GCN vs. RNN confirmed that the graph structure is indeed advantageous compared to the conventional sequence-based formulations. The authors also provided new results compared to the suggested baseline LIRPG and extended the comparison to continuous control tasks. I increased my score accordingly.

Correctness: Yes.

Clarity: Yes. Overall the paper is well written and easy to follow. The appendix is also informative.

Relation to Prior Work: There are some missing related papers. [a] Zheng, Zeyu, Junhyuk Oh, and Satinder Singh. "On learning intrinsic rewards for policy gradient methods." Advances in Neural Information Processing Systems. 2018. [b] Ibarz, B., Leike, J., Pohlen, T., Irving, G., Legg, S., & Amodei, D. (2018). Reward learning from human preferences and demonstrations in Atari. In Advances in neural information processing systems (pp. 8011-8023).

Reproducibility: Yes

Additional Feedback:

[Author Response · NeurIPS 2020]

Thank you for the constructive feedback. We are encouraged that the reviewers find our approach to be an interesting
way to encode the underlying graph [**R2**] and a scalable approach to solving more complex domains [**R3**, **R4**] that
results in considerable improvements [**R1**, **R2**, **R3**, **R4**] and compares well with existing methods [**R2**, **R3**, **R4**]. We
will address minor writing suggestions and incorporate the additional references. We now address some specific
questions and present a couple more results which will be included in the paper.

[**R1**,**R3**] **Time/computational complexity results.** We performed *additional analysis* in **Table 1** where we evaluated
each baseline on the Atari domain (other domains follow similar trend). We did these evaluations on a single V100 GPU,
8 CPUs and 40GB of RAM. The time taken (in frames-per-second (FPS), so high is good) for our approach $\Phi_{GCN}$ is
very similar to the PPO baseline, only slightly slower. We also compare favourably with respect to the RND, ICM,
LIRPG and Bi-LSTM [**R4**] baselines. We believe this good performance stems directly from our sampling strategy that
is minimal yet effective.

[**R2**,**R4**] **Continuous control.** Although continuous control presents challenges, our current algorithm, which relies
on sampling trajectories rather than constructing the full graph, is still an effective approach **as shown in additional**
**results for continuous environments provided below**. We conducted these experiments on the delayed Mujoco
domain where the extrinsic reward is rendered sparse by accumulating it over 20 steps before it is being provided to the
agent. We averaged the results over 10 random seeds. **Figure 1b-c** shows that our approach still provides significant
improvements over the PPO, LIRPG and Bi-LSTM baselines. In general, using graph-based learning in continuous
domains can be tackled in various ways, such as using grid cell-like constructs (which we discuss briefly in Sec.3.1),
or combine our sampling strategy with a model-based approach, in which we would roll out the model from states
observed on a trajectory. We will add more discussion on this to the future work section.

[**R4**] **On the advantage of GCN vs RNN.** In order to answer this question, **we performed additional experiments**
on the MiniWorld and Mujoco domains to verify whether a Bi-LSTM, together with the GCN's loss function, would
perform similarly. We chose a Bi-LSTM because it can propagate information both forward and backward in time,
which is better suited to our problem. In **Figure 1d** we see that although there is improvement over the PPO baseline,
the Bi-LSTM does not perform as well as the GCN based reward shaping. Moreover, in **Table 1** we notice that the
Bi-LSTM runs considerably slower than the PPO and GCN baseline. We believe that GCNs provide an advantage (even
for sampled trajectories) due to their architectural/structural bias, which has an important property: **local connectivity**.
In contrast, an RNN's output would depend on potentially all past states (in the case of LSTM/GRU this depends on the
weights themselves), and the bias is towards temporal connectivity on a particular trajectory, not local connectivity.
Because we essentially want to make predictions on the state space graph, local connectivity leads to better results. We
think a secondary factor is the fact that GCNs avoid exploding/vanishing gradients.

[**R1**] : **Inference or learning:** our paper focuses on both. Although $P(O|S)$ is clearly defined, we do not have access
to it since we do not have access to the MDP's reward function. We hope to clear this misunderstanding by moving
the algorithm box from appendix A.2 to the main paper. [**R1**] suggests that "it should not be as easy as stated in the
paper" but does not expand on this reasoning. We would like to argue that our sampling strategy is effective, scalable
and inexpensive as verified through various empirical evaluations (in the paper and in this rebuttal).

[**R3**,**R4**] **Related work and additional experiments:** We will gladly incorporate the suggested related works. Since
**LIRPG** is indeed a valuable baseline and has online code, **we performed additional experiments** on the same set
of 20 games from the Atari domain. In **Figure 1a** we plot the relative improvement with respect to PPO and see that
LIRPG achieves overall good but mixed results. In some environments it achieves good improvements, whereas in a
handful others the score is almost reduced to zero (note that our approach did not dramatically degrade performance).
An important issue related to LIRPG is its wall-clock time performance (in **Table 1**) which is a considerable roadblock
in terms of scalability and practicality.

| Method | FPS |
|---|---|
| PPO | 1126 |
| $\Phi_{GCN}$ | 1054 |
| Bi-LSTM | 815 |
| RND | 987 |
| ICM | 912 |
| LIRPG | 280 |

(a) LIRPG on Atari    (b) DelayHopper-v2    (c) DelayWalker2d-v2    (d) RNN reward shaping    Table 1: Frames-Per-Second (FPS) on Atari

Figure 1: Additional experiments

[Meta-Review · NeurIPS 2020]

The authors did a good job of addressing the concerns raised by the reviewers in their response. I hope that the authors will revise subsequent versions of the paper to reflect the clarifications they provided in their response and to address any lingering concerns in the reviews.